# Stress Management: Death Receptor Signalling and Cross-Talks with the Unfolded Protein Response in Cancer

**DOI:** 10.3390/cancers12051113

**Published:** 2020-04-29

**Authors:** Elodie Lafont

**Affiliations:** 1Inserm U1242, Université de Rennes, 35042 Rennes, France; elodie.lafont@inserm.fr; 2Centre de Lutte Contre le Cancer Eugène Marquis, 35042 Rennes, France

**Keywords:** death receptor, ER stress, unfolded protein response, TRAIL-R1/2, CD95, TNFR1, post-translational modifications, cell death, IRE1, PERK

## Abstract

Throughout tumour progression, tumour cells are exposed to various intense cellular stress conditions owing to intrinsic and extrinsic cues, to which some cells are remarkably able to adapt. Death Receptor (DR) signalling and the Unfolded Protein Response (UPR) are two stress responses that both regulate a plethora of outcomes, ranging from proliferation, differentiation, migration, cytokine production to the induction of cell death. Both signallings are major modulators of physiological tissue homeostasis and their dysregulation is involved in tumorigenesis and the metastastic process. The molecular determinants of the control between the different cellular outcomes induced by DR signalling and the UPR in tumour cells and their stroma and their consequences on tumorigenesis are starting to be unravelled. Herein, I summarize the main steps of DR signalling in relation to its cellular and pathophysiological roles in cancer. I then highlight how the UPR and DR signalling control common cellular outcomes and also cross-talk, providing potential opportunities to further understand the development of malignancies.

## 1. Introduction

Death Receptors (DRs) are a clade of transmembrane proteins belonging to the Tumour Necrosis Factor Receptor Super Family (TNFRSF). DRs comprise the broadly studied TNFR1, CD95/Fas/APO-1, TNF-Related Apoptosis-Inducing Ligand-Receptor 1 (TRAIL-R1)/DR4, and TRAIL-R2/DR5, on which we focus here. DRs respond to the binding by their cognate ligands, initiating cellular signals through Protein–Protein Interactions (PPIs). TNFR1, CD95 and TRAIL-R1/2 were initially shown to trigger cell death, including apoptosis, which involves caspase activation and is referred to as the extrinsic apoptotic pathway. This is achieved via the mandatory engagement of the cytoplasmic Death Domain (DD) [1,2]. TNF, CD95L and TRAIL, respectively bind TNFR1, CD95 and TRAIL-R1/2 amongst the DRs. In coordination with perforin-granzyme B [3,4,5], these ligand/receptor pairs contribute to the cytotoxic response of T and Natural Killer (NK) cells towards infected or tumour cells, thus participating to immunosurveillance [6,7,8,9,10,11]. Notably, DR participates in the killing potential of other immune cells like dendritic cells (DCs), macrophages and neutrophils. However, cell death induction is neither the sole nor necessarily the primary function of these DR, which can drive non-cytotoxic cellular outcomes, dependently or not on their DD, such as migration, differentiation, cytokine production and proliferation. Through these pleiotropic functions, DR participate to development and tissue homeostasis. In accord, the dysregulation of the tight control on both the induction of death and non-death functions by DR is implicated in certain inflammatory and auto-immune diseases as well as tumorigenesis. 

Akin to DR signalling, the response to Endoplasmic Reticulum (ER) stress is a master controller of cell/tissue homeostasis. ER stress conditions are common in tumours, owing to intrinsic (such as oncogene activation, high proliferation rate, secretory demand and/or mutational burden…) and extrinsic (including hypoxia, nutrients deprivation, acidic conditions …) cues [12]. Cellular adaptation to ER stress includes the activation of a pathway called the Unfolded Protein Response (UPR), initially aimed at restoring protein homeostasis. UPR can result in a plethora of cellular outcomes, including non-cytotoxic and cytotoxic ones and thereby influences tumorigenesis [13,14,15]. As detailed here, the pathophysiological relevance and the molecular bases of DR cellular outcomes are being deciphered, with several evolving complementary school of thoughts for their targeting in oncology. Furthermore, as developed here, the modalities of the cross-talk between DR signalling and the UPR are starting to be unravelled.

## 2. Cellular Roles and Molecular Determinants of DR Signalling and Current Directions for Its Targeting in Oncology

DRs fulfil both pro-tumorigenic and anti-tumour roles. On the molecular side, the initial ligand-binding induces the assembly of DR-associated protein complexes, a finely-tuned process that involves the DD and/or additional domains, like the Membrane Proximal Domain (MPD) [16]. A key difference in the first signalling steps within the DR bevy is the primary adaptor recruited through homotypic DD-mediated interactions, i.e., Fas-Associated protein with Death Domain (FADD) for TRAIL-R-1/2 and CD95 or TNFR1-Associated Death Domain protein (TRADD) for TNFR1. It is worth noting that a growing number of post-translational modifications (PTMs) of the ligands, DR as well as downstream signalling molecules—such as glycosylation [17], palmitoylation [18,19,20], nitrosylation [21], oxidation [22], phosphorylation [23], ubiquitination [24], cleavage [25,26,27,28]—majorly influence the aftermath of DR engagement and thus their pathophysiological roles. 

### 2.1. TNFR1 Signalling, Cellular Roles and Main Directions for Its Targeting in Oncology

As revealed early on, the systemic use of TNF induces a lethal shock syndrome [29], precluding the use of such an approach in oncology. Moreover, TNF preferentially induces gene-activation over cell death in vitro. On the molecular standpoint, gene-activation signal is initiated by the formation of a TNFR1-associated signalling complex (Complex I/TNFR1-SC), whereas cell death requires the formation of a secondary complex (Complex II) devoid of TNFR1 (Figure 1). Although the existence of these two platforms was defined more than 15 years ago by Micheau and Tschopp [30], new components and regulators of these are still being regularly identified [31,32,33]. TNF was also defined as an inducer of multiple cytokines [34,35] and its pro-inflammatory role was attributed early on to its preponderant ability to trigger gene-activation. This urged the design of TNF-neutralizing agents, like the TNFR2-Fc Enbrel (/Etanercept) or TNF-neutralizing antibodies, that are now successfully used worldwide to limit Rheumatoid Arthritis (RA), psoriasis and Inflammatory Bowel Disease (IBD) [36]. Nevertheless, some patients are primarily non-responders or become resistant to this therapy [37]. As explained in [38], strategies to target the pro-inflammatory role of TNF should also be considered in light of recent data demonstrating that aberrantly exacerbated TNF-induced death can also drive inflammatory phenotypes [39,40,41,42,43,44,45]. Moreover, recent data from the Walczak’s laboratory indicate that aberrant death initiated by TRAIL and CD95L contributes to inflammation too [46]. Combinatorial targeting of several DR could therefore become an option to treat cell death-driven inflammatory syndromes [38].

Studies in the 1990s demonstrated that perfusion of TNF in isolated limbs of patients presenting with melanoma or sarcoma induces anti-tumour effects with limited toxicity [47,48,49]. Thus, for some tumour contexts, fine-tuning TNF signalling might be beneficial. For example, molecular engineering to direct TNF towards tumour neo-vasculature can increase immune infiltration and impairs the growth of established melanoma and prostate tumours [50,51].

In line with the growing success of immunotherapy, triggering the TNF-induced Immunogenic Cell Death (ICD) of tumour cells is pursued with the view to enhancing Immune Checkpoint Inhibitor (ICI) efficiency. As reviewed in [59,60], combined antigenicity, adjuvanticity and inflammatory signalling characterises ICD. Beyond apoptosis, TNFR1, as well as CD95 and TRAIL-R1/2 [61,62,63,64], can elicit a programmed death called necroptosis [65]. Necroptosis arises upon inhibition of caspase activity (e.g., by specific viral proteins, high expression of the long isoform of cFLIP (cFLIP_L_) or pharmacological inhibition) and is mediated by the kinases RIPK1 and RIPK3 and the Mixed Lineage Kinase domain Like pseudo-kinase (MLKL). The latter oligomerizes, forming complexes of debated stoichiometry [66] that trigger plasma membrane permeabilisation. Like apoptosis, necroptosis is initiated by Complex II, the formation and killing-potential of which is controlled by multiple checkpoints (Figure 1). It is worth noting that RIPK3 expression is sometimes repressed in cancer cells through DNA methylation [67], thus strategies aimed at promoting necroptosis might also include hypomethylating agents. Cancer cells dying by necroptosis release inducible and constitutive immune-stimulatory molecules called Damage-Associated Molecular Patterns (DAMPs) [68] and display increased neo-antigens availability. The production of cytokines in necroptosis conditions has been observed [69], albeit lessened as compared to viable TNF-responding cells [70]. This is finely regulated, including at the MLKL level [71]. Interestingly, a surge in conventional (requiring the ER-Golgi secretory pathway) cytokine production at the early stages of TNF-induced necroptosis has been reported, prior to a decrease of this secretion at later time points [72]. Later cytokine production by necroptotic cells occurs even after plasma membrane permeabilization, provided that the integrity of the ER membrane is maintained, and could thus also contribute to immune cell recruitment [73]. Whether intact ER also contributes to the generation of other immune-modulators (such as metabolites) is unknown. Notably, multiple plasma membrane receptors are also shed, by A Disintegrin And metalloprotease (ADAM) family members, upon apoptosis and necroptosis induction and could thus contribute to the regulation of autocrine and paracrine signals in tumours [72,74]. NF-κB activation in dying cells was also reported to be crucial for effective T cell cross-priming [59]. Cancer cells dying by apoptosis can selectively release ‘find-me’ signals (such as nucleotides, cytokines or specific metabolites as further explored recently [75]) to promote the recruitment of phagocytes and subsequent engulfment. These cells can also display immunogenic properties in some contexts [59,60,76]. Checkpoints controlling TNFR1-induced cell death are largely mediated by PTMs, including ubiquitination (Figure 1). cIAPs are E3-ligases that limit DR-induced cell death [63,77,78] and can be pharmacologically targeted with Smac Mimetics (SM) [79,80,81]. In addition to their direct role in TNFR1 signalling, cIAPs constitutively limit non-canonical NF-κB activation and ensuing production of cytokines (including TNF itself). Therefore, lowering the threshold of TNF-induced death in tumours, through the targeting of cIAPs (together or not with TRAF2, another death-limiting factor [64,82,83,84]), enhances ICI efficiency [85,86].

In parallel, blocking TNF signalling is a tactic actively followed. Indeed, TNF can trigger pro-tumour functions, including tumour cell proliferation, invasiveness, Epithelial to Mesenchymal Transition (EMT), angiogenesis, the recruitment of tumour-promoting and the elimination of tumour-suppressive immune cells [87,88,89,90,91,92]. Considering this latter immune-modulatory role of TNF, combining TNF blocker and ICI has been tested. For instance, the Ségui’s laboratory reported that TNFR1 mediates the Activation-Induced Cell Death (AICD) of CD8+ melanoma-specific tumour infiltrating lymphocytes (TILs) [93] and that combining anti-TNF and anti-PD-1 promotes the regression of murine melanoma [94]. In accord, a phase 1b trial (NCT03293784) is ongoing combining anti-TNF and ICI for treatment of metastatic melanoma [95]. This combination might be beneficial for patients suffering from other types of cancer (for example colon cancer), including via the ability of TNF blockers to reduce some adverse effects of ICI [96].

Overall, rationale supports both blocking TNFR1 signalling or fine-tuning it for future anti-cancer therapies. In both cases, further research is needed to understand the mechanisms controlling DR signalling outcomes and its consequences on the tumour–stroma cross-talk. Defining clinically usable predictive markers (e.g., with regards to tumour immune status, the level of crucial cell death regulators as well as markers of additional stress pathways that influence cellular response, see part 2) will likely help to select patients which could benefit from one strategy over the other.

### 2.2. CD95 Signalling, Cellular Roles and Main Directions for Its Targeting in Oncology

Owing to the hepatic toxicity of CD95- agonistic antibodies [97], CD95 was first disregarded as a target in oncology. However, studying the CD95 signal prevailed when phenotypes linked to its defects were identified. Patients bearing germline and/or somatic mutations of CD95—or in fewer cases of CD95L—can indeed develop Autoimmune Lympho-Proliferative Syndromes (ALPS) [98]. This rare disease is characterized by pathognomonic lymphoproliferation sometimes with splenomegaly and hepatomegaly, the accumulation of circulating and lymphoid double negative (CD4− CD8−) T cells, hypergammaglobulinemia and autoimmune cytopenia [99,100]. In ALPS, CD95 mutations are mostly heterozygous and within its DD. Whereas these impair apoptosis, some non-cytotoxic pathways like Extracellular signal-Regulated Kinases (ERK) activation [101] are permitted. ALPS-like symptoms are observed in mice bearing mutations in CD95 *(lpr* or *lprcg* mice) or CD95L (*gld* mice), and, with enhanced auto-immunity, in CD95/CD95L KO mice [102,103]. The loss/re-expression of CD95L/CD95 in specific immune subsets (CD4, CD8 T or B lymphocytes or DCs) nicely highlighted the contribution of these cells in lymphoproliferation and auto-immunity [104]. Notably, CD95’s role in immune homeostasis might not solely be due to apoptosis but also to alternative roles, including PhosphoInositides 3-kinases (PI3K)/Akt activation [20,105,106].

ALPS patients have an increased risk of lymphoma [107] and *Lpr* mice develop lymphoma faster than controls when crossed with Eμ-Myc transgenic mice [108]. In mice, T cells, through CD95L, limit the spontaneous development of diffuse large B cell lymphoma (DLBCL) [109]. Together with CD95’s role as a mediator of immune cells cytotoxicity, this argues for a potential anti-tumour role of this DR. Since the hepatotoxicity of some CD95 agonists was attributed to an antibody dependent cell-mediated cytotoxicity [110], non-antibody based CD95 agonists were developed. As such, APO010, an hexameric CD95L fusion (two CD95L extracellular domain trimers fused to the collagen domain of adiponectin) [111] displayed some efficiency in glioma models [112,113]. Detailed results from a clinical trial evaluating its tolerability and efficiency in patients with solid tumours (NCT00437736) are not available yet. However, when considering CD95 agonists as a single treatment, caution is warranted beyond the risk of hepatotoxicity since these might also drive tumour-promoting signals.

CD95 can also fulfil oncogenic and immunosuppressive roles. For instance, CD95 loss limits tumour incidence in KRAS^G12D+/−^/PTEN^−/−^-driven ovarian cancer and diethlynitrosamine (DEN)-induced hepatocellular carcinoma (HCC) models [114]. Long-term CD95L stimulation promotes the proliferation of a population with stem cell markers in a Death-Inducing Signalling Complex (DISC) [115]- and type-I interferon-dependent manner [116] in various cancer cell lines. CD95L also promotes the expression of EMT markers by Pancreatic Ductal Adenocarcinoma (PDAC) cells and impairing CD95L/CD95 interaction (through CD95-Fc) limits PDAC growth in vivo [117]. In inflammatory models, CD95L induces the recruitment of leukocytes to inflammatory sites, such as myeloid cells in spinal cord injury [118] or neutrophils in sepsis [119]. Cancer cells can produce cytokines upon CD95 engagement, including while dying [120], thus impacting on immune cell recruitment. In tumours, several stromal cells, like endothelial cells [121], Cancer-Associated Fibroblasts (CAF) [122] or polymorphonuclear myeloid-derived suppressor cells (PMN-MDSC) [123], can express mCD95L and eliminate CD95+ CD8+ TILs. In addition, CD95 can promote the invasion of tumour cells. For example, CD95L induces the invasion of K-Ras mutated colorectal cancer cells [124]. Mechanistically, CD95 cooperates with PDGFRβ to activate a phospholipase Cγ1/PIP2/cofilin pathway, which is not counteracted by LIM-Kinase LIMK in a K-Ras mutated context, thus stimulating the formation of cell protrusions [125]. It is worth noting that, in K-Ras wildtype colon cancer cells, CD95L can mediate senescence, in a caspase-dependent manner [126]. In primary glioma cells and glioblastoma cell lines CD95 induces migration by recruiting Yes in a caspase-independent manner, forming a protein complex initiating a PI3K/Akt/GSK3β pathway, which promotes Matrix MetealloProteinases (MMPs) up-regulation. Hence, in a syngeneic orthotopic model, the co-injection of glioma cells with a CD95L-neutralizing antibody reduces tumour invasion [127]. In accord, blocking CD95 signal is one approach developed, for example, with APG101, consisting of the extracellular part of CD95 fused to an Fc domain. APG101 in combination with radiotherapy shows promising pre-clinical and Phase II clinical trial (NCT01071837) results for glioblastoma treatment [128,129]. Aberrantly increased CD95-driven apoptosis of erythroid progenitors contributes to myelodysplastic syndromes (MDS), which are characterised by haematopoiesis defects and can evolve in acute myeloid leukemia. APG101 has thus been tested for MDS and showed some potency in Phase I trial [130]. When envisioning DR-blocking strategies, one should consider that cytotoxic signals seem coordinated for immunosurveillance, with CD95L being preferentially engaged by CD8 T lymphocytes upon weak T-cell receptor stimulation [4] and TRAIL being used over granzyme/perforin by NK cells for serial killing [5].

Beyond blocking CD95 signal, one strategy is to fine-tune it. As for TNFR1, the switch between the different outcomes of CD95 signalling is influenced by PTM. Targets of these PTM include CD95 itself, with glycosylation, S-nitrosylation and S-palmitoylation [17,18,19,20,131] affecting its stability and function. Indeed, CD95 is detected at the cell surface as monomers, but also as homo-dimers or -trimers, pre-associated through the extracellular Pre-Ligand Assembly domain (PLAD) that encompasses parts of the Cysteine Rich Domains (CRD), which increases its ability to signal apoptosis [132,133]. CD95 can also be tyrosine phosphorylated within its DD (at Y291) by Src family kinases [134], which promotes its pro-survival function. Cohesive behaviour can influence CD95 tyrosine phosphorylation and CD95L response of both tumour and healthy cells. Indeed, apoptosis is preferentially induced when cell-to-cell contact is reduced whereas CD95L-induced CD95 tyrosine phosphorylation and Ras/ERK and PI3K/Akt pathway activation and proliferation is observed for cell clusters [135]. The molecular mechanisms by which global tyrosine activity is increased in clusters remains to be addressed. CD95L can be modified too. For example, its cleavage by MMPs within the extracellular stalk region releases soluble forms (altogether referred to as sCD95L herein). sCD95L, like the membrane-embedded counterpart (mCD95L), interact with CD95 [136,137,138,139,140,141,142,143]. Increased sCD95L concentrations are detected in the serum of patients suffering from various pathologies, including ALPS, systemic lupus erythematosus (SLE), RA, osteoarthritis or a certain type of cancers like NK-lymphomas [136,144,145,146]. However, the exact form(s) accumulated (and thus site(s) cleaved) are undefined. sCD95L forms are usually not cytotoxic, which is likely due to their trimeric nature since high-order assembly of CD95L—at least hexameric—is required for CD95-mediated apoptosis [147]. Noteworthy, some PTMs of CD95L, e.g., oxidation [22], alter its cleavage availability. The physiological role of CD95L cleavage was highlighted through the generation of knock-in mice expressing either mCD95L or sCD95L. This showed that sCD95L drives SLE-like autoimmunity and tumorigenesis while mCD95L impinges both [25]. On the molecular standpoint, sCD95L compete with mCD95L for CD95 binding [147,148]. sCD95L also induces alternative signals and outcomes, like activation of the ERK and NF-κB pathways and chemotaxis [25,146]. Interestingly, inter-CD95L distance was proposed to influence the kinetic of apoptosis induction [135]. Whether the rare cytotoxic forms of sCD95L (such as upon cleavage by plasmin [149]) have a modified conformation allowing this optimal apoptotic-prone inter-CD95 distance remains unknown. Overall, a deeper understanding of the regulation of CD95 signalling is needed if efficient strategies beyond blocking CD95L/CD95 interaction are to be brought safely to the clinic.

### 2.3. TRAIL-R1/2 Signalling, Cellular Roles and the Main Directions for Its Targeting in Oncology

TRAIL can bind five receptors: TRAIL-R1, 2, 3 and 4 and the soluble receptor osteoprotegerin, albeit with a weaker affinity for the latter. Transmembrane TRAIL-R1 and 2, as well as the murine ortholog TRAIL-R, possess DD and mediate both apoptotic and non-apoptotic outcomes. TRAIL-R3 is glycosylphosphatidylinositol-anchored, whereas TRAIL-R4 has a truncated DD. Both were proposed to act as decoys, limiting TRAIL-R1/2-mediated apoptosis through competition for TRAIL binding or impairment of DISC formation via hetero-oligomerisation [150]. Reports highlight that TRAIL-R3/4 activate non-apoptotic pathways, like NF-κB or Akt/PI3K, by cooperating with TRAIL-R1/2, or independently thereof [150,151]. Further research is, however, needed to understand the signal and physiological importance of TRAIL-R3/4. TRAIL can be cleaved to release sTRAIL, which is suggested to be less potent to induce apoptosis, especially through TRAIL-R2 [152] and might drive migration [153].

TRAIL deletion exacerbates ALPS-like phenotype in *gld* mice [154], whereas TRAIL- or TRAIL-R-deficient mice do not present major auto-immune manifestations. These animals are more susceptible to the development of some malignancies. Indeed, TRAIL-deficient mice present increased spontaneous tumour development (mostly lymphomas), as well as lymphomas arising upon the deletion of one allele of p53 [155]. Moreover, TRAIL-R^+/−^ mice have a significantly decreased survival in the Eμ-Myc lymphoma model [156]. Furthermore TRAIL, particularly expressed by NK cells, contributes to tumour immunosurveillance [157,158]. Since agonists of TRAIL-R1/2 kill tumour cells while sparing normal cells [159,160], several were developed and tested in clinical trials for cancer treatment. However, these displayed disappointing anti-tumour activities since the agonistic activity and half-life of the agents initially selected were limited. Multiple pre-existing or acquired cell death resistance mechanisms, such as caspase-8 mutations, cFLIP or anti-apoptotic Bcl-2 family members’ overexpression, are also likely germane to these disappointing clinical results. Thus, several approaches to design TRAIL-R agonist-based therapies with improved pharmacological properties are underway (for reviews: [161,162,163,164]). TRAIL-R1 or -R2-specific agonistic antibodies have been developed too. Owing to evidence that TRAIL-R1 and 2 might fulfil different functions [16,152,153,165,166,167,168], specificity for one over the other might be meaningful. However, one weakness of this approach could be that some cancer cells might not express both receptors. Another one could be their limited ability to induce higher-order TRAIL-R-oligomers formation. Interestingly, the combination of antibody-based and recombinant TRAIL-R-agonists results in TRAIL-R trimers multimerization, formation of the TRAIL-R-associated DISC and thus displays pre-clinical synergistic effects [169,170]. TRAIL-R agonists might also be useful to eliminate tumour-supportive stromal cells, like MDSCs [171] or endothelial cells [172]. 

In addition to its role in immunosurveillance, TRAIL elicits pro-tumour roles, the molecular bases of which are being uncovered. For example, certain tumour cells respond to TRAIL by proliferating, migrating and/or producing cytokines rather than dying [173,174]. Cytokine production by tumour cells upon TRAIL stimulation, which heavily relies on NF-κB [165,175], can induce chemotaxis [176,177] and thus participate to immune-modulatory functions, e.g., the recruitment of MDSCs [176]. On the molecular side, cytokine production is ignited by the formation of a TRAIL-R-associated Complex I—long considered solely as a DISC—and a TRAIL-R-devoid Complex II, which can both trigger death-inducing and gene-activation signals [178] (Figure 2). Both FADD and caspase-8, the latter acting as a scaffold [179], are crucial for TRAIL-induced gene-activation signalling and ensuing cytokine production [176,178,180]. The requirement of RIPK1 in TRAIL- and CD95L-induced gene-activation signalling seems to depend on the cell-type [176,178,179,181]. In some cell types, TRADD and RIPK1 are redundant for TRAIL/CD95L-induced gene activation and ensuing cytokine production [182]. TRADD is sometimes detected in a caspase-8-containing complex upon TRAIL stimulation, a phenomenon exacerbated upon RIPK1 deletion [179]. It is worth noting that, in Non-Small Cell Lung Cancer (NSCLC) lines, which resist to TRAIL-induced death, TRAIL can instead promote migration via a RIPK1-Src-STAT3 pathway [183]. TRAIL can also contribute to the elimination of activated T lymphocytes. Therefore, limiting TRAIL-induced death could also promote tumour immunosurveillance in some cases. Of note, TRAIL-R signal can emanate from intracellular compartments (also see part 2.2), influencing tumorigenesis. For example, the Trauzold’s laboratory highlighted that, in PDAC cells, nuclear TRAIL-R2 limits the processing of let-7 miRNA and promotes proliferation [184].

TRAIL-R1/2 apoptotic and non-apoptotic signals are influenced by PTMs, including ubiquitination [24]. For instance, cullin-3-mediated ubiquitination promotes caspase-8 activation [194], whereas LUBAC-mediated ubiquitination of RIPK1 and caspase-8 limits their activation and promotes NF-κB activation [178] upon TRAIL stimulation. Notably, some PTMs of DR signalling components can be triggered by pathogens [195,196,197], modulating the host immune response. The oncogenic context also influences tumour cells response to TRAIL (and CD95L). For example, the Wajant’s laboratory reported that mutant PI3Kα confers protection towards CD95L and TRAIL-induced apoptosis, downstream of caspase-8 activation and promotes amoeboid-like migration [198]. TRAIL and CD95L also trigger invasion of K-Ras mutated colorectal cells [124]. In K-Ras mutated PDAC and NSCLC, cancer cell-autonomous TRAIL-R signalling triggers proliferation, migration and invasion, through an MPD-mediated recruitment of Rac-1 and activation of PI3K. This results in increased tumour growth, metastasis and decreased survival in PDAC and NSCLC murine models [16]. 

Overall, understanding the precise mechanisms by which TRAIL-R impact on tumorigenesis is needed. This includes (i) defining the resistance mechanisms to TRAIL-induced cell death in patients’ tumours to select those that could benefit from TRAIL-R-targeting therapies and propose efficient combination therapies to circumvent resistance and (ii) understanding the pro-oncogenic role of TRAIL-R, including its impact on the cross-talk between tumour and stromal cells, and (iii) identifying markers (like KRas mutation) determining whether a TRAIL-R-blocking or-activating strategy would be beneficial. Considering the cross-talks between the Unfolded Protein Response (UPR) and DR signalling (see part 2.2) will likely be helpful in these different quests.

## 3. Cross-Talks between the UPR and DR Signalling in Cancer and Potential Vulnerability Points

ER stress response, which comprises the UPR, is induced by various conditions. It reshapes the RNA and protein composition through the modulation of transcription, translation, as well as the induction of RNA and protein degradation (the latter by autophagy or by ER-Associated protein Degradation, ERAD). The UPR primarily acts towards the recovery of protein homeostasis. The UPR, in particular, results in the expression of factors involved in protein folding and quality control and it also promotes the expansion of the ER itself. When this adaptive UPR fails—because ER stress is too intense or prolonged—UPR can induce cell death, with apoptotic or necroptotic features, a process then called terminal UPR. Aberrant UPR is associated with some human diseases, such as neurodegenerative diseases, diabetes, inflammatory diseases as well as cancer. Intriguingly, markers of the UPR are often constitutively activated in tumours, (like in Triple Negative Breast Cancer, TNBC, [199]), suggestive of a defective UPR-mediated cell death as well as a sustained subversion of the UPR towards tumour-supportive functions. Moreover, the UPR not only contributes to tumorigenesis but also to chemoresistance (for recent reviews: [12,13,14,15]). 

### 3.1. Main Actors of the Unfolded Protein Response

On the molecular standpoint, initiation of the UPR involves the engagement of three ER-resident sensors which are transmembrane proteins named Activating Transcription Factor 6 (ATF6), Protein kinase RNA-like ER Kinase (PERK) and Inositol-Requiring Enzyme-1 α/β (IRE1 α/β) (Figure 3). IREα (hereafter referred to as IRE1) is ubiquitously expressed when IRE1 β expression is found in the gastrointestinal and respiratory tracts. The activation of each sensor results from the detachment of its luminal domain from the chaperone Binding immunoglobulin Protein (BiP) [200,201], which recognises unfolded/misfolded proteins in the ER lumen. Both PERK and IRE1 can also directly bind to unfolded proteins, promoting their dimerisation and activation [202,203,204]. 

Oligomerised IRE1 undergoes a trans-autophosphorylation and a conformational switch. Indeed, IRE possess both a serine/threonine kinase activity and an endoribonuclease RNase activity on its cytosolic part. The IRE1 RNase cooperates with the tRNA ligase Rtcb to splice and re-ligate X-box Binding Protein 1 (XBP1) mRNA [205], thereby generating XBP1s mRNA. XBP1s is a transcription factor driving the expression of numerous proteins aimed at restoring ER homeostasis (such as oxido-reductases, foldases, proteins involved in ERAD or lipid metabolism) and promoting cell survival. Noteworthy, XBP1s can heterodimerize with other transcription factors, including ATF6 [206], influencing the selection of target genes. Upon unresolved ER stress, IRE1 nuclease activity can shift towards increased Regulated IRE1-Dependent Decay of RNA (RIDD) [207], which mediates the degradation of various RNAs (including mRNAs, miRNAs and rRNAs). Targeted mRNAs code for diverse proteins involved in cell survival, cell death, proliferation, migration and metabolism [208]. As developed in part 2.2, one RIDD target is TRAIL-R2, which has a dual role towards cell survival. RIDD can also promote cytokine production and cell death through the degradation of miR-17 and ensuing thioredoxin-interacting protein (TXNIP) up-regulation [209]. It is worth noting that a basal level of RIDD, in the absence of ER stress-induction, also exists, so RIDD activity is increased with ER stress duration/intensity [208] and is generally viewed as a pro-death activity in the context of terminal UPR. Notably, constitutive activation of IRE1-XBP1 is observed in TNBC cells and contributes to cancer stem cell expansion and chemoresistance [199,210]. The molecular means controlling whether IRE1 endoribonuclease activity preferentially mediates XBP1 mRNA splicing or the RIDD, and the extent and kinetics of their activation, are important regulators of cellular outcomes and are thus actively being studied. Evidence already highlights that IRE1 phosphorylation and oligomerisation magnitude influences the RIDD over XBP1-splicing activity engagement, even though the extent of oligomerisation required for each is debated [211,212,213,214,215]. IRE1 stabilisation [216], kinase activity and oligomerization are influenced by multiple PPIs and PTMs (for review: [217]). Additional PPI and PTM events downstream of IRE1, e.g., of XBP1 itself, also influence this shift. For example, the RIDD target protein-tyrosine phosphatase 1B (PTP1B) can dephosphorylate Rtcb, resulting in increased Rtcb activity and interaction with IRE1, thus providing a potential mechanism by which increased RIDD activity itself would counteract XBP1 splicing [218]. IRE1 can also exert scaffold functions through the recruitment of multiple proteins, the nature of which is still being defined, forming a ’UPRosome’. For example, TRAF2 can be recruited by IRE1 and initiate the NF-κB and c-Jun N-terminal Kinase (JNK) pathways [219,220]. The JNK activation downstream of TRAF2 and Apoptosis Signal-regulating Kinase 1 (ASK1) has been reported to contribute to death induction. For instance, JNK can phosphorylate and thereby inactivate Bcl-2 [221]. As reviewed recently [222,223], IRE1, including through the RIDD [224] and NF-κB [225], also differentially impacts on the anti-tumour immune response. Interestingly, a recent study by the Hetz laboratory highlights that IRE1 acts as a scaffold, binding filamin A [226] and promoting cell migration, whilst the implication of such a phenomenon in a cancer context remains to be investigated [14]. Furthermore, IRE1 also acts as a scaffold at mitochondria-associated membranes to dock inositol-1,4,5-triphosphate receptors, promoting mitochondrial calcium uptake, mitochondrial respiration and ATP production in resting as well as in ER stress conditions [227].

Similar to IRE1, PERK dimerizes and trans-autophosphorylates upon ER stress [229]. PERK up-regulates the transcription factor Nrf2 through ATF4 [230] and phosphorylates it [231,232], leading, in particular, to the expression of proteins aimed at restoring redox homeostasis. PERK also phosphorylates eIF2α, at S51, leading to an overall decrease in translation but also to an increase in specific protein translation, e.g., of ATF4. Of note, protein translation decrease also contributes to NF-κB activation [233]. ATF4 expression in particular leads to the up-regulation of proteins involved in redox and amino acid transport as well as the C/EBP-HOmologous Protein (CHOP/GADD153) transcription factor, which promotes the expression of various genes, including Bim [234], TRAIL-R2 (see part 2.2), and represses other (such as Bcl-2 [235]), thus contributing to death induction. Indeed, the intrinsic mitochondrial pathway also plays a role per se in ER-stress-induced death, whilst the members from the Bcl-2 bevy involved depend on the cell type. For example, the Strasser’s laboratory demonstrated early on that Bim is required for ER-stress-induced cell death in multiple cell lines and highlighted that ER stress activates Bim through Protein Phosphatase 2A (PP2A)-mediated dephosphorylation, preventing Bim degradative ubiquitination; and through CHOP-dependent transcription [234]. Three other kinases activated upon cellular stress -Heme-regulated eIF2α kinase (HRI) upon heme depletion or mitochondrial dysfunction [236,237]; Protein Kinase R (PKR), upon viral infection; and General Control Nonderepressible 2 (GCN2), upon amino acid deprivation- can phosphorylate eIF2α [238]. Thus, these kinases might link these selection pressure conditions with mitochondrial and/or DR signalling engagement. eIF2α phosphorylation is counteracted by the phosphatases complexes GADD34/PP1 and CReP/PP1. It is worth noting that GADD34 is a CHOP-target gene, providing a safeguard feedback mechanism during ER stress to limit CHOP expression and restore translation [239,240,241]. Interestingly, hyperactivation of the PERK/eIF2α branch has been described in human mammary epithelial cells undergoing EMT, and various mesenchymal breast cancer cell lines in which the hyperactivation of this branch might constitute a point of vulnerability [242]. It is also worth noting that the PERK/CHOP pathway is constitutively active in PDAC disseminated cancer cells and contributes to their immune evasion and metastatic dissemination [243]. Whether TRAIL-R signalling could be involved in these functions is intriguing. 

Once freed from BiP, ATF6 translocates to the Golgi apparatus [201], where it is cleaved by the proteases S1P and S2P, releasing a cytosolic fragment ATF6f, a bZIP transcription factor [244,245]. ATF6f-responsive genes include chaperones and proteins involved in ERAD. Of note, ATF6 also promotes the expression of XBP1(u) [246,247], thus cooperating with the IRE1 branch for XBP1s generation and ATF6 can also induce the expression of CHOP.

### 3.2. Cross-Talk between the UPR and DR Signalling: Molecular Mechanisms and Impacts on Cellular Outcomes

The UPR and DR signaling not only potentially act in parallel to control common cellular outcomes (cf part 1 and 2.1) but are also serially functionally linked at different levels. The existence of a link between the UPR and DR signalling was largely highlighted for the induction of cell death. Yet, recent data demonstrate that it also impacts on non-cytotoxic outcomes.

From the mid-1990s, multiple reports demonstrated that ER stress-inducers could trigger the death of tumour cells through the activation of the mitochondrial apoptotic pathway together with, or independently of, the DR pathway, depending on the cell type [248]. This has been mainly described for the link between the PERK branch and TRAIL-R signalling (Figure 4). Indeed, ATF4 leads to the up-regulation of CHOP, which can cooperate with other transcription factors, like c-jun [249], and bind to the TRAIL-R1/2 gene promoters to induce their expression in response to various ER stressors (for example glucose deprivation or treatments with classically used chemicals like thapsigargin (TG), tunicamycin (TN), Brefeldin A (BfA)). TG inhibits the sarco/endoplasmic reticulum Ca^2+^ ATPase (SERCA), thus leads to ER Ca^2+^ depletion, impacting on calcium-dependent ER chaperones and activates the UPR. TN, as an inhibitor of N-glycosylation, also activates the UPR. BfA inhibits the activity of ADP Ribosylation Factor (ARF) Guanine Exchange Factor (GEF) and consequently inhibits ER-Golgi vesicular trafficking. BfA leads to Golgi collapse, activation of the UPR and Golgi stress response (for review: [250]). TG can induce an up-regulation of TRAIL-R2, in some cases accompanied by an up-regulation of TRAIL-R1, and/or TRAIL [251] in different cancer cell lines [252]. The up-regulation of TRAIL-R2 upon ER stress has been attributed to both increased transcription as well as decreased degradation of its mRNA [251]. CHOP-dependent TRAIL-R2 up-regulation and ensuing TRAIL-R2-mediated cell death has been observed in multiple human cancer cell lines, such as colon cancer, prostate cancer [252], as well as normal cells under specific stress conditions (e.g., lipotoxicity in hepatocytes, [253]). The involvement of TRAIL-R2 and requirement of caspase-8 in TG-induced cell death has also been recently confirmed in Hct116 [254]. In several cancer cell lines [153], TRAIL-R1 also contributes, along with TRAIL-R2 to TG, TN or BfA-induced cell death. The up-regulation and intracellular accumulation of TRAIL-R1/2 has also been observed upon induction of Golgi stress and participate to the induction of cell death [255]. BfA treatment can promote the formation of a ‘DISC-like’ entity, containing TRAIL-R1/2, FADD and caspase-8 [153], likely at the ER or the ER-Golgi Intermediate Compartment (ERGIC). Notably, in melanoma cell lines, IRE1, and, at later stimulation time points both ATF6 and CHOP have also been reported to promote the expression of TRAIL-R2 upon TN treatment [256]. Glucose deprivation, to which cancer cells are particularly sensitive, induces ER stress and FADD/Caspase-8-dependent apoptosis in HeLa cells through the upregulation of TRAIL-R1 and 2 downstream of ATF4 and CHOP [257]. Similar to its role in CD95L/TRAIL-induced apoptosis, the engagement of the mitochondrial pathway through Bid is also required in certain cells, like Hct116, to mediate ER-stress-induced TRAIL-R2-dependent cell death [258]. In TNBC cell lines, blocking both TRAIL-R2 and the intrinsic pathway is required to prevent TG-induced apoptosis [259].

While concomitant upregulation of TRAIL-R1/R2, together with TRAIL, can logically lead to apoptosis, how the mere intracellular accumulation of TRAIL-R1/2, in the absence of TRAIL initiate death has long been puzzling. This was especially enigmatic considering that the resting-state unliganded TRAIL-R2 displays an autoinhibitory conformation. Indeed, contrary to CD95’s PLAD, in absence of TRAIL, the TRAIL-R2 PLAD counteracts homo-oligomerisation, which is mediated by the transmembrane helix [260]. This ‘self-inhibitory’ effect is relieved upon ligand binding. In line with this, disulfide bond-disrupting agents, which impact on the conformation of multiple proteins, were recently suggested to promote cancer cell death via ER-stress-induced up-regulation of TRAIL-R2 combined with increased TRAIL-R2 oligomerisation state, thus likely bypassing the PLAD-mediated inhibition [261]. Both the Ashkenazi’s and Walter’s laboratories recently provided an attractive explanation for the conundrum regarding the ligand-independent intracellular activation of TRAIL-R2 upon ER stress. Indeed, Pan et al. showed that TRAIL-R2 accumulates at the ERGIC upon ER stress and is able to directly bind ectopically expressed unfolded proteins. Hence, up-regulated TRAIL-R2 would constitute a late ER/Golgi stress sensor, which, through the recruitment of FADD, initiates apoptosis [262] (Figure 4). Whether or not additional DR (like TRAIL-R1) also fulfil the same function, remains to be addressed. Similarly, the impact of potential hetero-oligomers formation (such as those described between TRAIL-R2 and TRAIL-R4 [151,263], or with other TNFRSF members, like CD40 [264]) is unexplored in this context. Lastly, whether the modes of regulation described in part 1 for plasma-membrane-ignited DR-induced signalling, also apply to this ERGIC-initiated pathway remains to be explored and could offer targetable vulnerabilities point.

The mRNA encoding for TRAIL-R2 is an established target of the RIDD [265], and this activity of IRE1 can therefore limit TRAIL-R2 cell death, but also potentially non-death, signalling upon ER stress. Upon persistent ER stress, IRE1 and ATF6 activities were reported to be attenuated, contrary to the PERK branch that is maintained [266]. Recently, it was proposed that the phosphatase RNA Polymerase II Associated Protein 2 (RPAP2) dephosphorylates and thereby inhibits IRE1 RNase activity upon persistent ER stress [267]. Whilst it remains to be determined how RPAP2 and PERK act in the same pathway, this could constitute another checkpoint to ensure that ER stress, if prolonged, would induce sufficient up-regulation of TRAIL-R2 to promote cell death.

An interesting recent study by the Martin’s laboratory highlighted that TG, TN, BfA, as well as taxanes can induce cytokine (e.g., IL8, IL6, CXCL1) production through the UPR by triggering ATF4/CHOP-dependent TRAIL-R2 expression. TRAIL-R2, independently of TRAIL, then elicits a FADD/RIPK1/Caspase-8 pathway resulting in NF-κB activation [268]. Interestingly, part of the upregulated TRAIL-R2 was found intracellularly, suggesting that the gene-activation signalling might also be ignited from an ER/ERGIC TRAIL-R platform. Whether additional regulatory events, including PTMs (as detailed in part 1.3), also control this signalling remains to be defined. Very interestingly, recent work from the Muñoz-Pinedo laboratory highlights that multiple cancer cell types produce chemokines and cytokines (such as IL6, IL-8…) in an ATF4- and NF-κB-dependent manner in response to starvation, inducing the chemotaxis of macrophages, B cells and neutrophils [269]. Whether DR signalling is also involved in this case would be interesting to determine.

The modulation of the UPR can reciprocally affect exogenous TRAIL-induced signalling. For instance, Salubrinal, an inhibitor of eiF2α phosphatase complexes, increases TRAIL-induced apoptosis, via CHOP-mediated up-regulation of Bim in the HepG2 HCC cell line [270]. In human melanoma cell lines, TG-induced TRAIL-R2 up-regulation potentiates TRAIL-induced cell death [271]. Similarly, TN enhances CHOP-mediated TRAIL-R2 expression and sensitizes PC-3 prostate cancer cells to TRAIL-induced apoptosis [272]. In Mouse Embryonic Fibroblasts (MEFs), it has been shown that TN-induced sensitization to TRAIL could also be associated with a reduced ability of N-glycosylated mTRAILR to bind TRAIL and form Complex I [273]. On the contrary, in different murine cell types (L929, B16) and human cells, the N-glycosylation of mTRAIL-R, or TRAIL-R1, respectively, rather increased TRAIL-induced death [274]. In MDA-MB-231, TN induces a PERK-dependent but CHOP-independent up-regulation of TRAIL-R2. Along with a down-regulation of cFLIP_L/S_ and the anti-apoptotic Bcl-2 family member Mcl-1, this sensitizes MDA-MB-231 to TRAIL-induced apoptosis [275]. Nelfinavir, a Human Immunodeficiency Virus (HIV) protease inhibitor which also possess multiple anti-tumour properties [276], has been shown to bypass PERK activation and increases eIF2α phosphorylation by inhibiting CReP/PP1C [277]. Interestingly, Nelfinavir induces an ATF4/CHOP-dependent expression of TRAIL-R2, thus sensitizing GBM cell lines to TRAIL-induced apoptosis [278]. Whether these ER stress stimuli also impact on ligand-dependent DR-induced non-cytotoxic signalling remains to be addressed.

Since stromal cells are submitted to some of the same stress cues as tumour cells, the formers also sometimes display increased UPR activation, which impact on their proliferation, activation and function. Furthermore, modulation of the UPR can also impact on the tumour stroma. For example, Song et al. reported that ovarian cancer ascites limit glucose uptake and thereby lead to a high IRE1-XBP1 activation in CD4+ infiltrating T cells, which impairs their metabolism, activation and anti-tumour function [279]. The Glimcher’s laboratory recently highlighted that IRE1-XBP1 signalling in NK cells contributes to their expansion in infection contexts, as well as in a melanoma model [280]. Interestingly, in tumour-infiltrating MDSCs, UPR activation has been suggested to lead to an up-regulation of TRAIL-R2, which could actually constitute a point of vulnerability to target these tumour-promoting cells [281]. These different observations should thus be kept in mind to design UPR-, together or not with DR-, targeting strategies adapted to a specific tumour context, and again point to the necessity of identifying markers to orient the choice of therapeutics.

Functional links between ER stress and TNFR1 signals have also been documented (Figure 4). Of note, one study highlighted that CD95 expression is induced in macrophages upon ER stress [282]. IRE1, through the recruitment of TRAF2 and activation of IKKα/β, can initiate IκBα degradation and ensuing NF-κB activation. This leads to TNF production, which, at least in certain cell types (like MCF7), could participate to induction of cell death. Furthermore, pre-treatment with ER stress inducers (e.g., TN or TG) was reported to limit TNF-induced IκBα degradation and JNK phosphorylation and to sensitise cells to TNF-induced death. The ER-stress-induced depletion of TRAF2, resulting from increased protein degradation through an undefined mechanism, was proposed to account for this phenomenon [220]. TNF expression has also been observed downstream of IRE1/XBP1s in macrophages upon TLR2/4 engagement [283]. In MEFs, TNFR1 and RIPK1 were reported to form a complex with IRE1 to mediate ER-stress-induced JNK activation [284]. However, solely, the association of IRE1 and RIPK1, but not TNFR1, was observed in a later study in MEFs as well [285] and the requirement of TNFR1 or RIPK1 in JNK activation also appeared to be variable. Nevertheless, these studies shed light on a potential role of RIPK1 in UPR-induced death. In [285], RIPK1, independently of its catalytic activity, delayed caspase-8 activation and apoptosis induction in response to TN, TG or BfA. Moreover, RIPK1 deficiency did not impact on XBP1 splicing, CHOP induction, or JNK phosphorylation upon ER stress. The knock-down of TRAILR, TNFR1, CD95 or FADD did not majorly impact on TN-induced death. Whilst this remains to be demonstrated, the authors suggested that RIPK1, as a scaffold, impacts on IRE1 oligomerisation/activity to indirectly control death induction. The nature of cell death induced by ER-stress might be quite plastic, at least in some cell types. Indeed, in murine fibrosarcoma L929 cells, TNFR1, likely from an intracellular compartment, mediates ER-stress-induced necroptosis independently of TNF and this death can be switched to apoptosis upon RIPK1 depletion and back to necroptosis upon further caspase inhibition [286]. Intriguingly, IKKβ, in addition to mediating NF-κB activation, can also phosphorylate XBP1s, preventing its proteasomal degradation, in response to TNF stimulation [287], thus linking UPR and TNF signalling. Patients with heterozygous dominant mutations of TNFR1 suffer from a rare autoinflammatory disease termed TNFR1-Associated Periodic fevers Syndrome (TRAPS) [288]. TNFR1 mutations found in TRAPS have been reported to promote TNFR1 intracellular, reportedly at the ER, accumulation. Those patients present recurrent episodes of fever, soft tissue inflammation, peritonitis and sometimes amyloidosis and treatments usually include non-steroidal anti-inflammatory molecules, corticosteroids and IL1/IL1R-targeting molecules. Interestingly, TNFR1 mutants expressed in TRAPS are still able to initiate cell death and activate NF-κΒ and cytokine production [288,289,290], reminiscent of the role of TRAIL-R2 signalling from the ERGIC described above, which might be common to additional DR.

Several regulators of DR signalling are also logically shown to control ER-stress-induced death. In line with its role in controlling caspase-8 activation, cFLIP can limit ER-stress-induced apoptosis (e.g., in TNBC cell lines [259]). It is worth noting that the activation of the p65 unit of NF-κB, can counteract the ER-stress-induced expression of CHOP and could thus constitute a manner for some cells to circumvent ER-stress-induced apoptosis [291]. Prior or concomitant engagement of DR signalling upon ER stress might also counteract cell death, and should thus be controlled towards cell death induction. Of note, both transcriptional and translational up-regulation of cIAP1 [292], cIAP2 and/or XIAP has been observed upon ER stress conditions (for instance cIAP1 and 2 in NIH3T3 or MEFS, downstream of PERK/eIF2α [293], cIAP2 and XIAP in MCF7, downstream of Akt [294]), limiting death. Thus, the targeting of cIAPs using SM has been tested upon the induction of ER stress, in view to promote cell death. SM indeed sensitizes cancer cell lines to various ER stress inducers (Tg, BfA, dithiothreitol, but surprisingly not TM) [295]. Whether constitutive activation of UPR branches, as seen in tumours, impacts on SM response remains unknown. 

It is worth noting that, components of DR signalling and/or of the mitochondrial apoptotic pathway might impact on the intensity and/or kinetic of activation of the UPR. For example, Hetz et al. showed early on that Bax and Bak interact with and promote IRE1 activation [296]. Lately, Shemorry et al. reported that IRE1 can be cleaved in a caspase-dependent manner upon various ER stress conditions in heamatopoietic cancer cell lines and that this leads to the accumulation of an IRE1 ER-Lumenal Domain and TransMembrane segment (LDTM). The overexpression of the LTDM limits mitochondrial apoptotic pathway activation by impacting on Bax recruitment to the mitochondria [297]. Whilst this study needs to be followed up on (including by investigating how common this phenomenon is in non-haematopoietic cancer cells, defining the response of non-cleavable mutant-expressing cells, or the exact inhibition mechanism on Bax mitochondrial relocation, etc.), it does suggest an attractive DR-influenced checkpoint impacting on ER-stress-induced death.

## 4. Conclusions

UPR and DR signalling molecular mechanisms are being unravelled. These pathways can act both in parallel and serially to control common cellular outcomes, such as the induction of death, differentiation, cytokine production or migration. Whilst the molecular bases of their cross-talks are just starting to be defined, further understanding these will allow for the appropriate co-targeting of the UPR and DR signalling to synergistically impact on tumour cell fate, but also potentially on the dialogue between tumour cells and the stroma. Considering the tumour population, the hijacking of the UPR or DR signalling might co-occur in certain cells, but also constitute individual vulnerability points of distinct clones/subsets. Further work will thus be particularly needed to define the specific tumour contexts, and corresponding markers, for which such a co-targeting will be meaningful.

## Figures and Tables

**Figure 1 cancers-12-01113-f001:**
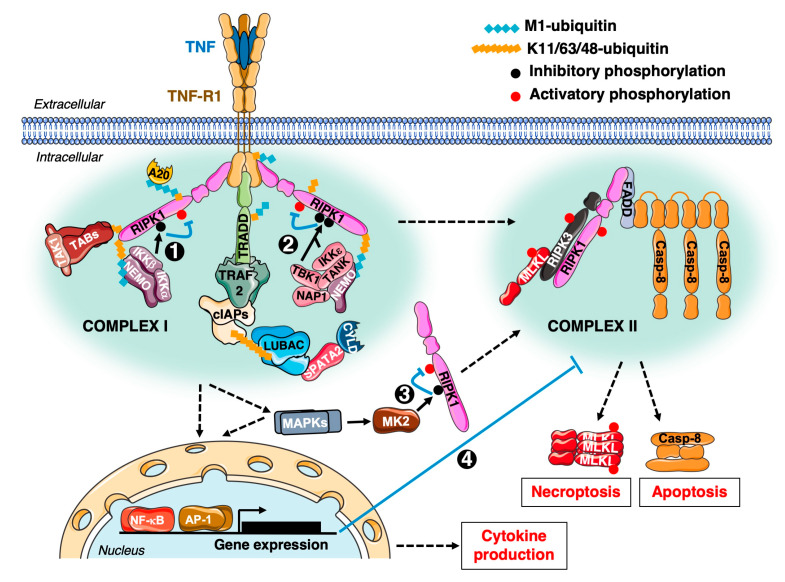
Tumor Necrosis Factor (TNF)/TNF-Receptor 1 (TNFR1)-mediated signalling and cell death checkpoints. Following binding of TNF to TNFR1, two complexes can be assembled: the TNFR1-associated Complex I (gene-activation) and a TNFR1-devoid Complex II (cell death induction) [30]. Complex I formation starts with the recruitment of TNFR1-Associated Death Domain protein (TRADD) and Receptor-Interacting serine/threonine Kinase 1 (RIPK1) to the Death Domain (DD) of TNFR1. TRADD further recruits TNFR-Associated Factor 2 (TRAF2) and cellular Inhibitor of Apoptosis Protein (cIAP)1/2. cIAP1/2 poly-ubiquitinate (ubiquitin linked through K11, K48 or K63) several components of Complex I. A recent report indicates that cIAP1, through K48 ubiquitination of RIPK1, promotes the degradation of the latter, thus limiting induction of cell death [52]. cIAP1/2-formed chains also recruit the Linear UBiquitin chain Assembly Complex (LUBAC), which further modifies several components with M1-linked ubiquitin, stabilizing Complex I. The absence of cIAPs and/or LUBAC results in reduced ubiquitination in, and stability of, Complex I and promotes the formation of Complex II. In complex I, cIAPs- and LUBAC-formed chains recruit the TGF-Beta Activated Kinase 1 (TAK1)/TAK1-Binding Proteins (TAB) complex and several Nuclear Factor-κB (NF-κB) Essential Modulator (NEMO)-containing complexes. The TAB/TAK complex activates Mitogen-Activated Protein Kinase (MAPK) pathways and allows for the activation of the NF-κB pathway by NEMO/Inhibitor of NF-κB Kinase α (IKKα)/IKKβ. IKKα/β, IKKε and TANK-Binding Kinase 1 (TBK1) directly phosphorylate RIPK1 [32,33,53,54] and thereby inhibit its autoactivation (checkpoints #1 and #2). This impedes RIPK1 detachment from Complex I and thus Complex II formation. A similar function is fulfilled by the p38 MAPK target MK2 [55,56,57] in the cytosol (checkpoint # 3). Downstream TNF-target genes include deubiquitinases (like CYLD and A20), pro-inflammatory cytokines, as well as proteins like cellular FLICE-Like Inhibitory Protein (cFLIP) which inhibit cell death initiation (cf part 1.3) at the level of complex II (checkpoint # 4) [58]. Whilst not represented here, some ubiquitinated RIPK1 and RIPK3 have also been detected in Complex II.

**Figure 2 cancers-12-01113-f002:**
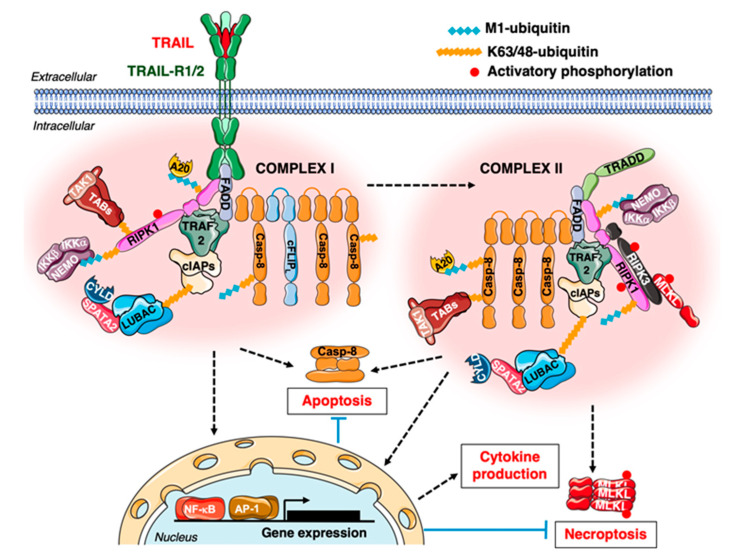
TNF-Related Apoptosis-Inducing Ligand (TRAIL)/TRAIL-Receptor 1 and 2 (TRAIL-R1 and 2)-mediated death and gene-activation signalling. Following binding by TRAIL, TRAIL-R1/2 recruit the adaptor Fas-Associated protein with Death Domain (FADD), initiating the formation of Complex I (initially termed the DISC [185,186]). In turn, FADD interacts, through its Death Effector Domain (DED), with pro-caspase-8, inducing its oligomerisation as chains [187,188,189]. This activates caspase-8 with cleavage steps separating the tandem DED and the catalytic subunits that are released to form activated caspase-8. The latter cleave the effector caspases 3/6/7, which, by cleaving hundreds of substrates, induce apoptosis. Activated caspase-8 also cleaves Bid, producing t-Bid, which links Death Receptor and mitochondrial apoptotic pathways, bolstering effector caspases activation (not shown, detailed in [164]). In certain cell types (like hepatocytes), caspase-8 activation is limited and efficient effector caspases activation requires mitochondrial permeabilization, releasing Second Mitochondria-derived Activator of Caspases (SMAC), which, through antagonism of the caspase-inhibitor X-linked IAP (XIAP), activates effector caspases [190]. In Complex I, cFLIP long, short or related (L/S/R) isoforms are also recruited via DED-mediated protein-protein interactions. cFLIP_S/R_ isoforms solely display DEDs and abrogate caspase-8 activation (not shown). cFLIP_L_ possesses, in addition to DEDs, a large and a small subunit similar to caspase-8, but is inactive as it lacks a catalytic cysteine. Depending on its expression level, cFLIP_L_ promotes or limits caspase-8 activation [191,192,193]. Importantly, the cFLIP_L_/caspase-8 heterodimer, also present in Complexes II of TRAIL and TNF signalling, can cleave substrates in its vicinity. These include RIPK1 and RIPK3, thus cFLIP_L_/caspase-8 limit necroptosis. Several E3-ligases (like cIAP1/2, cullin-3, LUBAC) are recruited to Complex I and ubiquitinate components therein, regulating caspase-8 activation and/or recruiting the gene-activation TABs/TAK1 and NEMO/IKKα/IKKβ complexes [24,63,83,178,194]. As a consequence, Complex I can initiate gene-activation too. By undefined mechanisms a Complex II, devoid of TRAIL-R1/2, can be formed. This complex was long viewed as mediating gene activation [180], yet it also signals cell death. A single Complex I and Complex II are depicted, but different types of complexes—with varying stoichiometry of the components depicted— likely co-exist in a given cell, at a given stimulation timepoint.

**Figure 3 cancers-12-01113-f003:**
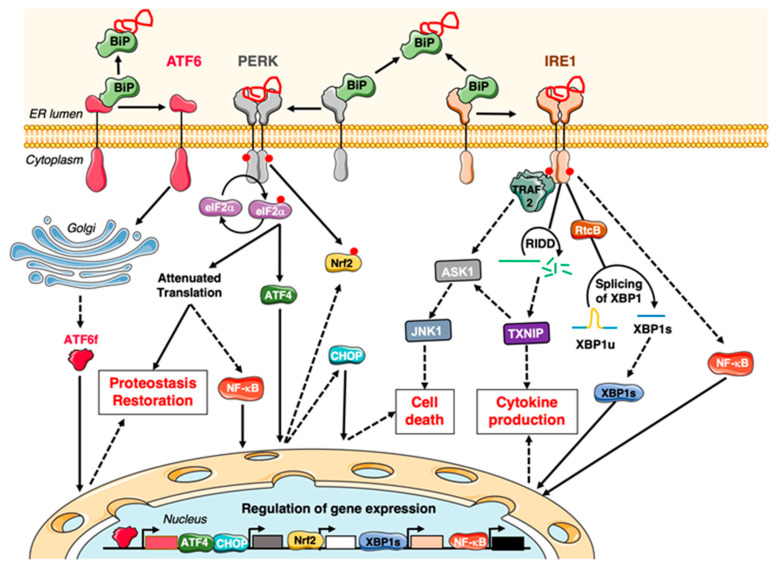
The Unfolded Protein Response (UPR). Each branch and sub-branch of the UPR impacts on multiple cellular outcomes (only some of which are represented, see text). Beyond the relative intensity of activation of each branch, their dynamic of activation also plays a role in controlling cell fate [228].

**Figure 4 cancers-12-01113-f004:**
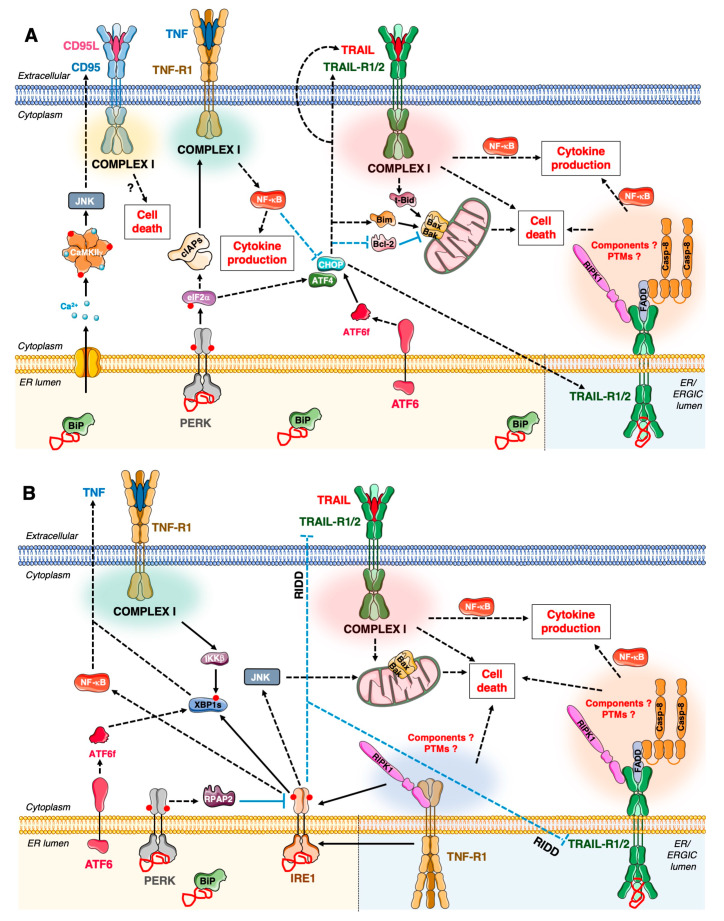
Functional links between DR signalling and UPR. **A**. Focus on the Protein kinase RNA-like Endoplasmic Reticulum Kinase (PERK)-DR signalling cross-talks. TRAIL-R1/2 contribute to ER-stress-induced death, including via the mitochondrial branch in certain cell types, as well as ER-stress-induced cytokine production. Whether or not it is also involved in additional UPR-mediated signalling outcome is not known. TRAIL-R1/2 accumulate at the ER or the ER-Golgi Intermediate Compartment (ERGIC) membrane upon ER stress, from which they mediate signalling. TRAIL-R2 oligomerisation in these compartments was recently proposed to be mediated by direct unfolded protein binding. The composition of the TRAILR-associated complex(es) in these intracellular compartments and the mode of regulation of this signal remain to be investigated. **B**. Focus on the Inositol-Requiring Enzyme-1 (IRE1)-DR signalling cross-talks. Through the RIDD, IRE1 represses TRAILR-dependent signalling. IRE1 activity is regulated by multiple mechanisms including its putative interactions with TNFR1, RIPK1, and via RNA Polymerase II Associated Protein 2 (RPAP2)-mediated dephosphorylation. (See text for further details on depicted cross-talks and Figure 1, Figure 2 and Figure 3 for details on Complexes 1 and initiation of UPR signalling.)

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
