# Peer review of "Stress Management: Death Receptor Signalling and Cross-Talks with the Unfolded Protein Response in Cancer"

_cancers, 2020, doi:10.3390/cancers12051113_

Round 1

Reviewer 1 Report

This is an interesting review exploring cross-talk between DR and UPR pathways. The article is structured so as to first explore distinct components of DR signalling with indication of directions for targeting in cancer, i.e. TNFR1, CD95, TRAIL-R1/2. Subsequently, the author discusses cross-talks between UPR and DR signaling, and this is followed then  by the most interesting part of the article where mechanisms of cross-talk between UPR and DR are described (3.2). This part focuses especially on TRAIL-R2, and further highlights RIPK1.

This is a very thorough and well-written review exploring known and new directions at the UPR/DR interface.  However, Figure 4 is disappointing; it is essentially Figure 3, a UPR representation, with on the right-hand side representation of TRAIL-R signaling and the nexus between both NF-kB, this does not reflect the text. My advice is  to schematize the multiplex intersections of TRAIL-R2 with the UPR/ER stress pathways as discussed in the main text .

Reviewer 2 Report

This is a very comprehensive and useful review on the cellular effects of ttwo signaling systems: the UPR and the Death Receptor with emphasis Trail R1/2.  The key and more innovative aspect of this review is the cross-talk between the two signaling pathways. This is an emerging field and this review covers  a large body of literature  represents therefore a useful reference for others interested in the field.

The 4 figures captures well the topic and the multiple interactions.
